# ESBL-producing *Klebsiella pneumoniae* in a University hospital: Molecular features, diffusion of epidemic clones and evaluation of cross-transmission

**Pierre Edwige L. Fils**[1], **Pascal Cholley**[1,2], **Houssein Gbaguidi-Haore**[1,2], **Didier Hocquet**[1,2,3], **Marlène Sauget**[1,2,3]*, **Xavier Bertrand**[1,2]

**1** Hygiène Hospitalière, Centre Hospitalier Universitaire, Besançon, France, **2** UMR-CNRS 6249 Chrono-environnement, Université de Bourgogne Franche-Comté, Besançon, France, **3** Centre de Ressources Biologiques—Filière Microbiologique de Besançon, Centre Hospitalier Régional Universitaire, Besançon, France

* msauget@chu-besancon.fr

**Data Availability Statement:** All relevant data are within the paper.

## Abstract

The worldwide spread of *Klebsiella pneumoniae* producing extended-spectrum β-lactamase (ESBL-Kp) is a significant threat. Specifically, various pandemic clones of ESBL-Kp are involved in hospital outbreaks and caused serious infections. In that context, we assessed the phenotypic and molecular features of a collection of ESBL-Kp isolates in a French university hospital and evaluated the occurrence of potential cross-transmissions. Over a 2-year period (2017–2018), 204 non-duplicate isolates of ESBL-Kp were isolated from clinical (n = 118, 57.8%) or screening (n = 86, 42.2%) sample cultures. These isolates were predominantly resistant to cotrimoxazole (88.8%) and ofloxacin (82.8%) but remained susceptible to imipenem (99.3%) and amikacin (93.8%). CTX-M-15 was the most frequent ESBL identified (83.6%). Multilocus sequence typing and pulse-field gel electrophoresis analysis showed an important genetic variability with 41 sequence types (ST) and 50 pulsotypes identified, and the over representation of the international epidemic clones ST307 and ST405. An epidemiological link attesting probable cross-transmission has been identified for 16 patients clustered in 4 groups during the study period. In conclusion, we showed here the dissemination of pandemic clones of ESBL-Kp in our hospital on a background of clonal diversity.

## Introduction

Over the four past decades, the worldwide spread of extended-spectrum β-lactamases *Enterobacterales* has become a significant threat [1–3]. The recent emergence of carbapenem-resistant *Enterobacterales* has further restricted antimicrobial treatment options and has amplified the threat to public health [4] Carbapenem-resistant and ESBL-producing *Enterobacterales* are in the WHO priority pathogens list for research and development of new antibiotics [5].

**Funding:** This work was supported by a grant from the University of Franche-Comté to P.E.L.F.

**Competing interests:** The authors have declared that no competing interests exist.

Among *Enterobacterales*, *K. pneumoniae* producing extended-spectrum β-lactamases (ESBL-Kp) is an important nosocomial pathogen with the potential to cause serious infectious diseases such as bacteremia and pneumonia [6–8]. Recently multi-drug-resistant *K. pneumoniae* emergence has led to incurable infections [9–12]. Several class of ESBLs have been described in *K. pneumoniae*. ESBLs that derived from penicillinases TEM and SHV emerged in the 1980s and CTX-M type enzymes such as CTX-M-15 have arisen during the 2000s [13]. This change reflects an important capacity for gene transfer, possibly between epidemic clones [14–16]. The pandemic clones ST258, ST11, ST15, and ST147 spread since two decades and recently, the CTX-M-15-producing ST307 clone emerged globally [17]. Most hospital outbreaks are due to this multidrug-resistant *K. pneumoniae* clones [17–19]. The emergence of carbapenem-resistant strains further complicates the management of these infections [4, 10, 20].

In France, *K. pneumoniae* is the 5[th] most prevalent pathogen responsible for healthcare-associated infection [21]. In the last 15 years, the proportion of *K. pneumoniae* isolates from bloodstream infections resistant to third-generation cephalosporins increased from 4% (in 2005) to 29% (in 2017) with more than 80% of this resistance being due to ESBL production [22]. On the other hand the proportion of carbapenem-resistant *K. pneumoniae* was to 1% in 2019 [23]. Thus, in view of the wide diffusion of epidemic clones, our objective was to describe the phenotypic and molecular characteristics of the ESBL-Kp isolated among patients hospitalized in the Besançon University hospital in 2017 and 2018. Secondarily, we have evaluated the cross-transmission of ESBL-Kp major clonal groups within our hospital.

## Materials and methods

### Setting, study period and patients included

We conducted a retrospective cohort study for a 2-year period (from January 2017 to December 2018) in the Besançon University hospital, a 1400-bed hospital with approximately 50,000 admissions and 320,000 patient-days annually. Over the study period, each hospitalized patient with at least one clinical isolate or one screening isolate positive with ESBL-Kp was included. Day care admissions and consultations were excluded. Screening is only carried out in specific hospitalization ward with particular risk, including medical and surgical intensive care units, haematology, neurosurgery and nephrology units. During the study period 10,637 ESBL screenings were performed for 4,238 patients.

### Bacterial isolates

All the isolates were identified as *K. pneumoniae* by MALDI-TOF MS Microflex LT (Bruker Daltonik GmbH, Bremen, Germany) according to the manufacturer's recommendations and routinely tested for ESBL production using the synergy test [24]. ESBL-Kp isolates were stored at—80˚C at the Centre de Ressources Biologiques Filière Microbiologique, Besançon (CRB-FMB, Biobanque BB-0033-00090). For patients with multiple positive samples, we retained for further analysis only the first isolate of ESBL-Kp.

### Antibiotic susceptibility testing

The activity of 13 antibiotics (amoxicillin, amoxicillin/clavulanic acid, ticarcillin, ticarcillin/clavulanic acid, piperacillin/tazobactam, cefotaxime, cefoxitin, imipenem, ertapenem, ofloxacin, amikacin, and the combination sulfamethoxazole/trimethoprim) was assessed according to EUCAST recommendations [24].

## Molecular genotyping

To identify the genes encoding ESBL, the DNA of all the isolates were screened as described before [25]. For the identification of carbapenemases, all isolates non-susceptible to ertapenem (*i.e.* zone diameter $\geq$ 25 mm around a 10-μg ertapenem disk) were tested by PCR for the presence of $bla_{OXA-48}$, $bla_{KPC}$, and $bla_{NDM}$ genes [26]. The sequence type (ST) of isolates was determined by multilocus sequence typing (MLST) according to Diancourt scheme [27]. The clonality of all ESBL-Kp isolates was investigated by pulsed-field gel electrophoresis (PFGE) following *Xba*I digestion and pulsotypes (PTs) were defined according to international recommendations [28].

## Patient data collection

To evaluate the cross-transmission of epidemic clones, we consulted the medical records of patients carrying isolates displaying the major PTs of ST405 and ST307 clonal groups. We collected retrospectively from January 2017 to December 2018 the following data for each patient: (*i*) data about the current hospitalization (dates of admission and discharge, length of stay, hospitalization ward), (*ii*) date and type of sample positive with ESBL-Kp, and (*iii*) patient outcome. Cross-transmission was considered as probable between two patients if they were hospitalized in the same department over the same period of time.

## Ethics statement

The French regulation allows the study of bacterial strain along with their associated data after information of the patient. Approval of ethical committee was not required in that particular case. The patients whose anonymized data (age; risk factors) were given the following information: "Use of samples and microorganisms for research purposes: samples (blood samples, biopsies, surgical specimens) can be taken to establish a diagnosis and to adapt your treatment or that of your child. Some of these samples or the microorganisms they contain may be stored for diagnostic or research purposes. Your samples are anonymized. The medical data associated with the samples and the microorganisms are collected in a computer file authorized by the CNIL, Commission Nationale de l'Informatique et des Libertés (French Data Protection Authority). You have the right to access and rectify the data entered. You may at any time reconsider your decision without any consequences for your care (or that of your child) and oppose the use of biopsies and operating documents for research by contacting the Franche-Comté Regional Tumorotheque (Tel. +33 3 81 66 89 66) or oppose the use of blood samples or microorganisms for research by contacting the Centre de Ressource Biologique–Filière Microbiologique (Tel. +33 3 70 63 21 34).

## Results

Over the two-year survey, 204 non-duplicate ESBL-Kp isolates were retrieved from clinical (n = 118, 57.8%) or screening (n = 86, 42.2%) sample cultures. Among these, 146 (72%) isolates were available for further analysis. ESBL-Kp isolates came from rectal swab or feces (n = 59; 40.4%), urines (n = 54; 37.0%), blood (n = 9; 6.2%) and other samples (n = 24; 16.4%). The isolates were found mainly in the intensive care units (n = 49; 34%), followed by haematology (n = 25; 17%), emergency (n = 17; 12%), surgery (n = 11; 8%) and hepatology units (n = 7; 5%).

Our isolates were predominantly resistant to cefotaxime (98.0%), cefepime (90.4%), ofloxacin (82.8%) and to the combination sulfamethoxazole/trimetoprim (88.8%). However, a small

proportion of ESBL-Kp were resistant to carbapenems (0.7% and 6.8% for imipenem and ertapenem, respectively) and to amikacin (6.2%).

A large majority of the 146 ESBL-Kp isolates available for further analysis produced ESBLs of the CTX-M group, with 122 producing CTX-M-15 (83.6%), 5 producing CTX-M-14 (3.4%), 4 producing CTX-M-3 (2.7%), and 1 producing CTX-M-9 (0.7%). We found only 10 and 4 isolates producing SHV (6.9%) and TEM-like ESBLs (2.7%), respectively (Table 1). Among the 10 isolates non-susceptible to carbapenems, only one harbored a carbapenemase-encoding gene ($bla_{OXA-48}$).

Our ESBL-Kp collection was distributed in 41 STs among which ST405 (36 isolates) and ST307 (29 isolates) predominated. Others STs including more than 5 isolates were ST336 (10 isolates), ST15 (9 isolates), ST13 (5 isolates) (Table 1). PFGE analysis distributed 146 isolates in 50 PTs (from PT1 to PT50). The three predominant PTs (PT28, PT26, and PT38) regrouped isolates (35, 26, and 10) from three major STs (ST405, ST307, and ST336), respectively (Table 1).

Overall, we have identified two major clonal groups: (*i*) ST405 with the majority of isolates, (35 out of 36) that clustered in the PT28, (*ii*) ST307 with the majority of isolates (26 out of 29) that clustered in the PT26. All isolates of these dominant PTs carried $bla_{CTX-M-15}$ except one ST307 that harboured $bla_{CTX-M-3}$ (Table 1). Fig 1 represents the timeline of hospital stay of the patients positive with ESBL-Kp of ST405 and ST307 major PTs to evidence probable cross-transmission. For ST307, we identified three patient clusters over the study period: (*i*) one in hepatology implying three patients, (*ii*) one in the medical intensive care unit with 3 patients, and (*iii*) one in the surgical intensive care unit implying 5 patients (Fig 1A). Besides, five out of the eight patients (P12, P14, P16, P17, P19) positive with ESBL-Kp ST405 had an obvious epidemiological link in the haematology unit (Fig 1B).

**Table 1. Characteristics of ESBL-producing *K. pneumoniae* isolates in the University hospital of Besançon (eastern France), 2017–2018 (n = 146 non-duplicate isolates).**

| Sequence type | No of isolates | Pulsotype (no of isolates) | ESBL-encoding gene (no of isolates) | Proportion susceptible to the main antibiotics* (%) | | | | | | | |
|---|---|---|---|---|---|---|---|---|---|---|---|
| | | | | P | PI | ESC | FOX | CPE | OFLO | AN | SXT |
| ST405 | 36 | PT28 (35) | $bla_{CTX-M-15}$ | 0 | 6 | 0 | 92 | 99 | 14 | 97 | 6 |
| | | PT1 (1) | $bla_{CTX-M-15}$ | | | | | | | | |
| ST307 | 29 | PT26 (26) | $bla_{CTX-M-3}$ (1), $bla_{CTX-M-15}$ (25) | 0 | 0 | 0 | 77 | 95 | 0 | 100 | 4 |
| | | PT33 (2) | $bla_{CTX-M-15}$ | | | | | | | | |
| | | PT13 (1) | $bla_{CTX-M-15}$ | | | | | | | | |
| ST336 | 10 | PT38 (10) | $bla_{CTX-M-15}$ (10) | 0 | 0 | 0 | 78 | 94 | 0 | 80 | 0 |
| ST15 | 9 | PT5 (4) | $bla_{CTX-M-15}$ (4) | 0 | 26 | 0 | 71 | 89 | 0 | 100 | 44 |
| | | PT15 (2) | $bla_{CTX-M-15}$ (2) | | | | | | | | |
| | | PT32 (2) | $bla_{CTX-M-15}$ (2) | | | | | | | | |
| | | PT45 (1) | $bla_{CTX-M-15}$ (1) | | | | | | | | |
| ST13 | 5 | PT8 (5) | $bla_{CTX-M-15}$ (5) | 0 | 7 | 0 | 40 | 100 | 20 | 100 | 0 |
| Other STs (36) | 57 | 39 PTs | $bla_{CTX-M-3}$ (3), $bla_{CTX-M-9}$ (1), $bla_{CTX-M-14}$ (5), $bla_{CTX-M-15}$ (34), $bla_{SHV-2}$ (8), $bla_{TEM-like}$ (4), $bla_{SHV-11}$ (1), $bla_{SHV-12}$ (1) | 0 | 29 | 15 | 66 | 96 | 34 | 89 | 18 |

*P: penicillins (amoxicillin, ticarcillin), PI: penicillins with β-lactamase inhibitors (amoxicillin/clavulanic acid, piperacillin/tazobactam, ticarcillin/clavulanic acid), ESC: extended-spectrum cephalosporins (cefepime, cefotaxime), FOX: cefoxitin, CPE: carbapenems (ertapenem, imipenem), OFLO: ofloxacin, AN: amikacin, SXT: sulfamethoxazole/trimethoprim.

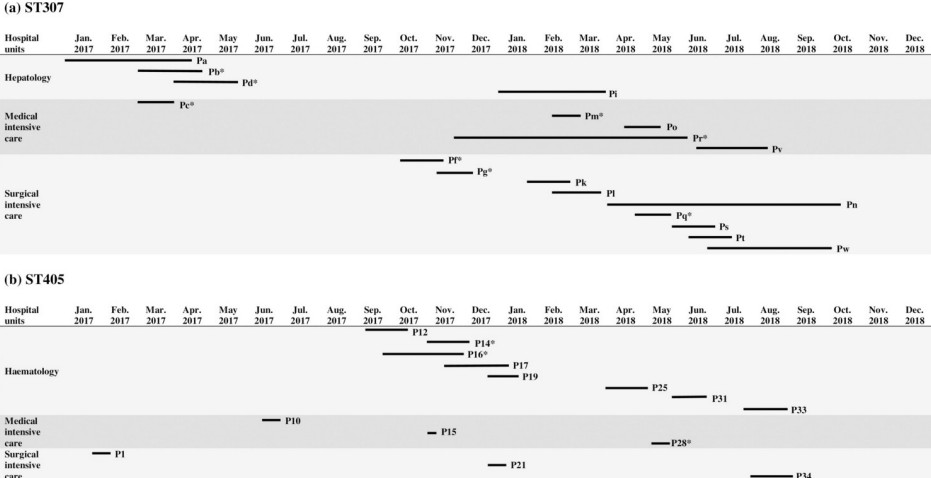

**Fig 1. Representation of the hospital stays of the patients carrying the major pulsotype of the two epidemic clones ST307 (A) and ST405 (B) of *K. pneumoniae* producing extended-spectrum β-lactamases within the Besancon University hospital, 2017–2018.** Each line represents a patient whose code is given at the right end of the line. Asterisk after patient code indicates his death. We only represent wards that have hosted ≥ 2 patients positive with ST307 ESBL-Kp (n = 18) and ST405 ESBL-Kp (n = 14).

## Discussion

ESBL-Kp are essentially transmitted in the hospital setting from patient to patient directly by the healthcare workers' hands or indirectly via the environment [15]. Although gastrointestinal tract of colonized patients is the main reservoir of hospital outbreaks, environmental sources of contamination such as medical devices and U-bends have also be incriminated [29–31]. Infection control measures to limit cross-transmission between patients can be implemented. They may be limited to standard precautions (SPs) which can be complemented by additional precautions contacts (APCs) and patient screening to prevent transmission. During our study period, an epidemiological link identified probable cross-transmission in 4 clusters implying 16 patients. Given the number of patients who shared identical PT, other cross-transmissions have undoubtedly been missed probably due to undetected carriers or acquisition from an environmental reservoir.

We isolated ESBL-kp mostly from patient hospitalized in intensive care and haematology units who benefit from a weekly systematic screening ESBL-producing *Enterobacterales*. As expected, CTX-M-15 was the most frequent ESBL enzyme. Genotyping revealed both the genetic variability of the ESBL-Kp collection with 50 PTs and 41 STs identified, and the over-representation of the epidemic clones ST405 and ST307. Recent work confirmed the emergence of the ST307 clone, frequently identified as being responsible for hospital outbreaks of ESBL-Kp [32]. A phylogenetic analysis based on genomic data dated the emergence the ST307 clone in the 1990s. Then, the clade that spread worldwide displayed mutations in *gyrA* and *parC* quinolone resistance determining regions that confer the resistance to fluoroquinolones and harbored a plasmid that contains $bla_{CTX-M-15}$ and the resistance determinants *sul2*, *dfrA14*, *strAB*, and the optional *aac(3)-IIa* [17]. The diffusion of this multidrug resistant clone is all the more worrying that ST307 strains producing carbapenemase such as KPC, VIM, OXA-48, and NDM have been recently described [33]. The propagation of this ST lineage in Italy and Korea has caused concern [20, 32, 34, 35]. In our study, although ST307 isolates were distributed into 3 PTs, the fact that one PT gathered nearly all the isolates was in favor of a large intra-hospital spread. In accordance with other studies, ST307 ESBL-Kp isolates from

our collection mostly harbored $bla_{\text{CTX-M-15}}$, they were ofloxacin and sulfamethoxazole/trimethoprim resistant (100% and 96%, respectively) but not to amikacin (0%). The only OXA-48 carbapenemase-producing ST307 isolate that we have identified belonged to one of the sporadic PT. The early identification of this carbapenemase-producing isolate triggered specific infection control measures that presumably avoid its spread.

The clustering of CTX-M-15 producing ST405 isolates into a dominant PT was also in favor of its intra-hospital diffusion. Other groups reported hospital epidemics with ESBL-Kp ST405 producing CTX-M-15 [8, 36, 37]. Although this clone has been reported to produce both CTX-M-15 and a carbapenemase, all but one ST405 isolate of our collection were susceptible to carbapenems [38–43]. Similarly, the global ST15 clone often produce both an ESBL and a carbapenemase [18, 19, 44, 45] while all ST15 isolates of our collection were susceptible to imipenem [8]. The overall proportion of antibiotic resistant isolates in the present collection is consistent with the literature and suspected outbreak isolates remained susceptible to cefoxitin, carbapenems and amikacin (Table 1) [46].

This study has some limitations. First, the ESBL-Kp were collected in a single hospital in France. A multi-centric study could determine whether the epidemic clones identified here (*i. e.* ST307, ST405) also spread in other French hospitals. Although the typing methods (*i.e.* MLST, PFGE) used here were appropriate to investigate local outbreaks of bacterial pathogens, the whole genome sequencing would have (*i*) provided more detailed information on ESBL-Kp phylogeny and (*ii*) allowed the identification of virulence genes and the genetic environment of the ESBL-encoding genes.

## Conclusion

We documented in a University Hospital (Eastern France) the dissemination of two epidemic clones of ESBL-kp (ST405, ST307) on a background of clonal diversity. Implementation of additional infection control measures for patients positive with ESBL-Kp is fully justified. Further studies are needed to identify the genetic and biological features that favor the spread of these multidrug resistant epidemic clones.

## Author Contributions

**Conceptualization:** Xavier Bertrand.

**Formal analysis:** Pierre Edwige L. Fils, Marlène Sauget.

**Funding acquisition:** Xavier Bertrand.

**Investigation:** Pierre Edwige L. Fils, Pascal Cholley.

**Methodology:** Houssein Gbaguidi-Haore.

**Project administration:** Xavier Bertrand.

**Resources:** Xavier Bertrand.

**Supervision:** Xavier Bertrand.

**Validation:** Marlène Sauget.

**Writing – original draft:** Marlène Sauget.

**Writing – review & editing:** Didier Hocquet, Xavier Bertrand.

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
