## [Decision Letter · Decision Letter 0]

15 Dec 2020

PONE-D-20-32043

ESBL-producing in a University hospital: Molecular features, diffusion of epidemic clones and evaluation of cross-transmission

PLOS ONE

Dear Dr. Sauget,

Thank you for submitting your manuscript to PLOS ONE. After careful consideration, we feel that it has merit but does not fully meet PLOS ONE’s publication criteria as it currently stands. Therefore, we invite you to submit a revised version of the manuscript that addresses the points raised during the review process.

We look forward to receiving your revised manuscript.

Kind regards,

Grzegorz Woźniakowski, Full professor, PhD, ScD

Academic Editor

PLOS ONE

Journal Requirements:

Reviewers' comments:

Reviewer's Responses to Questions

**Comments to the Author**

1. Is the manuscript technically sound, and do the data support the conclusions?

Reviewer #1: Yes

2. Has the statistical analysis been performed appropriately and rigorously? 

Reviewer #1: Yes

3. Have the authors made all data underlying the findings in their manuscript fully available?

Reviewer #1: Yes

4. Is the manuscript presented in an intelligible fashion and written in standard English?

Reviewer #1: Yes

5. Review Comments to the Author

Reviewer #1: Article "ESBL-producing Klebsiella pneumoniae in a University hospital: Molecular features, diffusion of epidemic clones and evaluation of cross-transmission" is interesting. It present problem of bacterial cross-transmission in one hospital in France. Total paper and all results are connected with one medical facility in France, it could be disadvantage but authors have handled with that issue well. For major concern is for me the introduction and statistics. Introduction is to short and must be extend for more information about current global and France situation with the K. pneumoniae. There is a lot of percent data but there is lack of statistics. It requires at lest confidence interval and p value. In case of minor errors I notice:

Line 26: Do not Begin abstract with the word “Purpose”

Line 40: do not use word “conclusion” twice in the one line.

Line 56: such as? Give some examples

Sincerely,

Reviewer

6. PLOS authors have the option to publish the peer review history of their article (what does this mean?). If published, this will include your full peer review and any attached files.

Reviewer #1: No

---

## [Author Response · Author response to Decision Letter 0]

21 Jan 2021

Dear Editorial Board Member,

We thank you for the review of our manuscript “ESBL-producing Klebsiella pneumoniae in a University hospital: Molecular features, diffusion of epidemic clones and evaluation of cross-transmission”. We are very grateful for the opportunity to address the reviewer concerns and comments, as well as some you raised. Below are our responses to these comments and questions. We hope that these responses address all of these comments, and we thank you and the reviewers for helping us to improve the quality of the manuscript.

Response to the reviewer

Reviewer #1: 

Comment 1: Introduction is to short and must be extend for more information about current global and France situation with the K. pneumoniae.

- Answer: Additional informations has been added in the 'introduction' section (lines 63-65 and lines 70-71). 

Comment 2: There is a lot of percent data but there is lack of statistics. It requires at least confidence interval and p value.

- Answer: We thank you for this advice but they are not applicable to our data. Although our study include many percent data these are only descriptive data without value comparisons or trend analysis. Our objective was only to describe the K. pneumoniae producing extended-spectrum β-lactamases population structure but not to present the evolution of this population over the study period.

Comment 3: In case of minor errors I notice: Line 26: Do not Begin abstract with the word “Purpose”. Line 40: do not use word “conclusion” twice in the one line.

- Answer: We agree with this comment. In the abstract, we have removed the paragraph headings (lines 26, 31 and 40).

Comment 4: Line 56: such as? Give some examples.

- Answer: Done line 56.

We hope that these responses address all of these comments, and we thank you and the reviewers for helping us to improve the quality of the manuscript.

Thank you again for your consideration,

Best regards, 

Marlène

---

## [Editor Report · Decision Letter 1]

16 Feb 2021

ESBL-producing Klebsiella pneumoniae in a University hospital: Molecular features, diffusion of epidemic clones and evaluation of cross-transmission

PONE-D-20-32043R1

Dear Dr. Sauget,

We’re pleased to inform you that your manuscript has been judged scientifically suitable for publication and will be formally accepted for publication once it meets all outstanding technical requirements.

Kind regards,

Grzegorz Woźniakowski, Full professor, PhD, ScD

Academic Editor

PLOS ONE
---

## [Editor Report · Acceptance letter]

5 Mar 2021

PONE-D-20-32043R1 

ESBL-producing *Klebsiella pneumoniae* in a University hospital:
Molecular features, diffusion of epidemic clones and evaluation of cross-transmission 

Dear Dr. Sauget:

I'm pleased to inform you that your manuscript has been deemed suitable for publication in PLOS ONE. Congratulations! Your manuscript is now with our production department. 

Kind regards, 

on behalf of

Prof. Grzegorz Woźniakowski 

Academic Editor

PLOS ONE